# A network analysis of problematic smartphone use in Japanese young adults

**Masaru Tateno** [1,2], **Takahiro A. Kato** [3], **Tomohiro Shirasaka** [4], **Junichiro Kanazawa** [5], **Wataru Ukai** [2], **Tomoya Hirota** [6]*

1 Department of Child and Adolescent Psychiatry, Tokiwa Child Development Center, Tokiwa Hospital, Miki, Japan, 2 Department of Neuropsychiatry, Graduate School of Medicine, Sapporo Medical University, Sapporo, Japan, 3 Department of Neuropsychiatry, Graduate School of Medical Sciences, Kyushu University, Fukuoka, Japan, 4 Department of Psychiatry, Teine Keijinkai Medical Center, Sapporo, Japan, 5 Department of Clinical Psychology, Health Sciences University of Hokkaido, School of Psychological Science, Tobetsu, Japan, 6 Department of Psychiatry and Behavioral Sciences, University of California San Francisco, San Francisco, California, United States of America

* tomoya.hirota@ucsf.edu

## Abstract

### Background

We aimed to explore the overall network structure of problematic smartphone use symptoms assessed by smartphone addiction scale-short version (SAS-SV) and to identify which items could play important roles in the network.

### Methods

487 college and university students filled out the study questionnaire, including SAS-SV. We constructed a regularized partial correlation network among the 10 items of SAS-SV. We calculated three indices of node centrality: strength, closeness, and betweenness, to quantify the importance of each SAS-SV item.

### Results

We identified 34 edges in the estimated network. In the given network, one item pertaining to withdrawal symptom hadthe highest strength and high closeness centrality. Additionally, one item related to preoccupation was also found to have high centrality indices.

### Conclusion

Our results indicating the central role of one withdrawal symptom and one preoccupation symptom in the symptom network of problematic smartphone use in young adults were in line with a previous study targeting school-age children. Longitudinal study designs are required to elicit the role of these central items on the formation and maintenance of this behavioral problem.

**Data Availability Statement:** All relevant data are within the paper and its Supporting information files.

**Funding:** This work was partially supported by the Grant-in-Aid for Scientific Research on (1) KAKENHI—the Japan Society for the Promotion of Science- JSPS (JP16H06403 & JP20H01773) and (2) The Japan Agency for Medical Research and Development - AMED (JP19dk0307073 & JP21wm0425010). The funders had no role in study design, data collection and analysis, in the decision to publish or the preparation of the manuscript. All the funding was received during this study. There was no additional external funding received for this study.

**Competing interests:** The authors have declared that no competing interests exist.

## Introduction

The penetration rate of the Internet in Japan has increased dramatically over the last two decades, from 9.1% in 1997 to 89.8% in 2019 [1]. One of the characteristic features of the latest changes in the Internet environment is the wide and rapid distribution of smartphones in all age groups, especially among young generations. In recent years, the number of internet users who access the Internet through a smartphone (63.3%) is higher than those who use the Internet via desktop and/or laptop computers (50.4%) [1].

Smartphones were launched in Japan in 2007, and three years later, the rate of smartphone users remained only 9.7% [1]. However, in accordance with the improvements in information and communication technology, the rate of smartphone users reached 83.4% in 2019. As the number of smartphone users becomes higher, problems related to smartphone overuse have become more serious.

As a historical standard, addiction is commonly referred to as, and associated with, substance abuse. However, the connotation of the term has been expanded since the 1990s to include behaviors that may lead to rewards (e.g., gambling, shopping, sex) [2]. In 2011, the American Society of Addiction Medicine (ASAM) released an official statement defining all addictions, including behavioral addictions, in terms of brain changes [3]. In the statement, ASAM provided a clear definition, namely that "Addiction is a major chronic disease of the brain's reward, motivation, memory, and related circuits." Although the diverse positions taken by various proponents have yet to culminate in a complete consensus [4, 5], in general, problematic smartphone use is considered as a subtype of behavioral addiction in which a person is excessively engaged in challenging behaviors that are not substance-related, even though they have a negative impact on the person's physical, social, economic and mental well-being [6, 7].

Problematic smartphone use is considered the result of excessive and ultimately destructive smartphone overuse and is characterized by clinical features of behavioral addiction; preoccupation, compulsive behavior, lack of control, functional impairment, withdrawal and tolerance [8, 9]. In response to the increase of smartphone users, studies regarding problematic smartphone use have been reported from various countries [10–12].

Several research groups developed questionnaires to screen for and assess problematic smartphone use, including the Smartphone Addiction Scale (SAS) [13–16]. The short version of SAS (SAS-SV) consists of 10 items selected by a consensus among experts in addiction psychiatry from the original version of SAS which is composed of 33 items [13, 17]. At present, SAS and SAS-SV are widely used self-rating scales to assess the severity of smartphone addiction in the world, and have been translated into several languages [18].

Historically, researchers have used a latent variable model, where the items (symptoms, signs) are manifestations of a particular underlying attribute (problematic smartphone use, for example) to understand the disease model. In this model, the observed items are independent of each other given an individual's score on the latent variable (local independence) [19]. However, behavioral addiction is a complex human behavior phenomenon, and is composed of heterogeneous observable signs and symptoms. Therefore, treating problematic smartphone use within a latent variable model may overshadow meaningful associations existing between individual symptoms.

Although not mutually exclusive, a network model has been recently gaining attention as an alternative approach to the latent variable model in understanding the psychopathology of mental disorders [20, 21], including behavioral addiction [22, 23]. In the network model, problematic smartphone use is not an underlying latent unobservable disease entity. Rather, it is considered to be a complex network of mutually reinforcing symptoms [24]. This approach

allows for identifying the overall network structure of disorders by quantifying associations among observable symptoms. Additionally, the network approach allows for identifying the most central symptom(s) that potentially play a critical role in the onset and maintenance of disorders. Two research groups have recently employed this approach to elucidate the network structure of problematic smartphone use in preadolescents and adolescents, where symptoms pertaining to withdrawal, loss of control, and continued excessive use were deemed as central symptoms [18, 25]. However, no network studies in this field have focused on individuals transitioning to adulthood, a population which is considered uniquely vulnerable to mental health disorders, including addiction problems [26]. Nearly everyone in this age group possesses smartphones with scant or no parental controls, and thus behaviors related to smartphone use in this population would not be directly influenced by family factors (parenting style, for example).

In this study, we aimed to: 1) explore the overall network structure of problematic smartphone use symptoms in Japanese college students, and 2) identify the most central symptoms in the network.

## Materials and methods

### Participants

The subjects of the study were 487 private college and university students in Sapporo city and its environs in Japan. Study participation rate was 81.2% (487/600). Their academic deviation scores were average or a little below the average compared to the national standard in Japan. Research collaborators for data collection were recruited through personal connections of the first author of this paper (MT). Nine teachers from three universities and six colleges agreed to voluntarily support our investigation. This study is a secondary data analysis using data obtained for a study focusing on internet addiction, smartphone addiction, and the Hikikomori trait that was conducted between July and October 2018 and was previously reported elsewhere [27]. Characteristics of the study participants are summarized in Table 1. In this study, the participants received the questionnaire sheets and filled out the questionnaire in the classroom.

**Table 1. Characteristics of the study participants.**

| | Whole (n = 487)<br>(Male 132, Female 355) |
|---|---|
| Age (mean ± SD) | 19.6±1.5 |
| SAS-SV (mean ± SD) | 29.6±8.8 |
| Internet use (hours) (mean ± SD) | |
| Weekdays | 4.86±3.1 |
| Weekend | 6.82±4.1 |
| Purpose (%) | |
| Gaming | 42 (8.6) |
| SNS | 303 (62.2) |
| Video-sharing | 101 (20.7) |
| Music | 23 (4.7) |
| Web searches | 13 (2.7) |
| Others | 5 (1.0) |

SAS-SV: Smartphone Addiction Scale–Short Version, SD: Standard Deviation, SNS: Social Network Service

## Measures

**Smartphone addiction scale–short version.** The original version of the SAS was developed as a self-report scale in South Korea [13]. The SAS includes 33 questions that assess six domains relating to smartphone overuse (continued overuse, loss of control, preoccupation, withdrawal, tolerance, and functional impairment) on a six-point Likert scale ranging from 1 (strongly disagree) to 6 (strongly agree). The Japanese-version of the SAS was developed, and its reliability and factor validity were assessed in 1,037 youth aged 15–24 years [28]. Kwon et al. also developed the short version of SAS (SAS-SV) by extracting 10 questions from the SAS to use it as a smartphone addiction screener and confirmed its validity [17]. In translating the SAS-SV into Japanese, our research team made a minor modification by inserting the term LINE [29] into the sentence of SA8 (Constantly checking my smartphone so as not to miss conversations between other people on Twitter or Facebook); however, the Japanese version of the SAS-SV has not yet been validated in Japanese youth.

The 10 items in the SAS-SV are listed in Table 2. The researchers in the present study discussed and defined the function of each SAS-SV item by referencing diagnostic criteria of gaming disorder found in the International Classification of Diseases, 11th Revision (ICD-11) and internet gaming disorder of Diagnostic and Statistical Manual of Mental Disorders, Fifth Edition (DSM-5) to assist readers in interpreting the findings of this study. The internal consistency of the Japanese-version SAS-SV for the participants in the present study was rated good, with Cronbach's alpha being 0.83.

## Data analyses

**Network estimation.** In the present study, we constructed a regularized partial correlation network using a Graphical Gaussian Model between the 10 symptoms of problematic smartphone use. In this network model, nodes represent individual items, and relationships between nodes are defined as edges. Edges are understood as partial associations between two nodes while controlling for all other nodes in the network, and their thickness or weights reflect the strength of association (an estimation of partial correlations coefficients). The lack of edge

**Table 2. Ten items of smartphone addiction scale—Short version and mean scores in this study.**

| | SAS-SV Items | Consensus among authors of this study | Mean scores (mean ± SD) (n = 487) |
|---|---|---|---|
| SA1 | Missing planned work due to smartphone use | functional impairment (occupational) | 3.5 ± 1.4 |
| SA2 | Having a hard time concentrating in class, while doing assignments, or while working due to smartphone use | functional impairment (academic) | 3.3 ± 1.4 |
| SA3 | Feeling pain in the wrists or at the back of the neck while using a smartphone | functional impairment (physical) | 2.3 ± 1.4 |
| SA4 | Won't be able to stand not having a smartphone | withdrawal | 3.1 ± 1.6 |
| SA5 | Feeling impatient and fretful when I am not holding my smartphone | withdrawal | 2.2 ± 1.3 |
| SA6 | Having my smartphone in my mind even when I am not using it | preoccupation | 2.1 ± 1.2 |
| SA7 | I will never give up using my smartphone even when my daily life is already greatly affected by it. | loss of control | 3.2 ± 1.5 |
| SA8 | Constantly checking my smartphone so as not to miss conversations between other people on LINE, Twitter or Facebook | loss of control | 2.9 ± 1.4 |
| SA9 | Using my smartphone longer than I had intended | continued overuse | 4.5 ± 1.2 |
| SA10 | The people around me tell me that I use my smartphone too much. | continued overuse | 2.5 ± 1.4 |

SAS-SV: Smartphone Addiction Scale–Short Version

SD: Standard Deviation

between two nodes means conditional independence relationships among the nodes. The network was estimated and visualized using the R-package 'qgraph' in the statistical program 'R version 4.0.3' [30, 31]. In order to create a more parsimonious network and minimize the likelihood of type-I errors, the graphical LASSO (Least Absolute Shrinkage and Selection Operator), reducing the edges by shrinking the smallest edges exactly to zero, in the R package qgraph. This process of regularization is coupled with best fit model selection, by minimizing an information criterion, in this case, Extended Bayesian Information Criterion (EBIC) [32]. We used the Fruchterman-Reingold algorithm for network visualization, a force-directed algorithm that encourages closely related nodes to be plotted near each other [33].

**Centrality.** We calculated three indices of node centrality to quantify the importance of each of the 10 symptoms in the SAS-SV network [34]: closeness (the average shortest path between a given node and the remaining nodes in the network); betweenness (the number of times that a node lies on the shortest path between two other nodes); and strength (the sum of the absolute value of its connections with other nodes in the network). Previous research showed strength to be the most robust centrality measure [35]. These centrality values are typically presented as standardized Z-scores, with higher values reflecting greater overall importance of a symptom to the network.

**Network stability.** To investigate the reliability and robustness of the study results, we assessed the accuracy and stability of network edges and centrality indices using the R package 'bootnet' [19]. More specifically, using nonparametric bootstrap methods, we estimated network stability as follows: 1) constructed a 95% bootstrapped confidence interval around the regularized edge weights, 2) computed an edge-weight difference test, and 3) estimated the correlation stability coefficient of centrality indices (via a case-dropping bootstrap procedure). Centrality indices were considered strongly stable if the values of the correlation stability coefficient were over 0.5, while values below 0.25 indicated inadequate stability [35].

**Ethical issues.** This study was approved by the ethics committee of Tokiwa Hospital. The study's aim was stated on the cover page of the questionnaire sheets that requested voluntary respondents to answer all questions anonymously. Answering the questions was deemed to constitute consent.

## Results

### Characteristics of the study participants

Table 1 shows the characteristics of the study participants (n = 487). 355 female students (mean years of age: 19.4, standard deviation (SD): 1.4) and 132 male students (mean years of age: 20.2, SD: 1.8) completed the questionnaire. More than half of participating students used the internet primarily for social networking services (SNSs) (female: 70%, male: 40.9%). The rate of gaming was higher in males compared to that in females, 18.9% and 4.8%, respectively.

### Network structure and centrality

Fig 1 depicts the network structure of relations among SA symptoms in study participants. We identified 34 edges in the estimated network, among which the weights of the edges between the items SA1 (Miss planned work due to smartphone use) and SA2 (Difficulty focusing on assignments/work due to smartphone use) and those between the items SA5 (Feel impatient and uneasy when not holding my smartphone) SA6 (Having my smartphone in my mind even when I am not using it) were large (0.49 and 0.41, respectively). Other strong associations include those between SA4 (Unable to stand not carrying a smartphone) and SA7 (Continued use of my smartphone despite its negative impact on daily life) (0.33) and between SA4 and SA5 (0.27). All edge weights of this network are listed in S1 File.

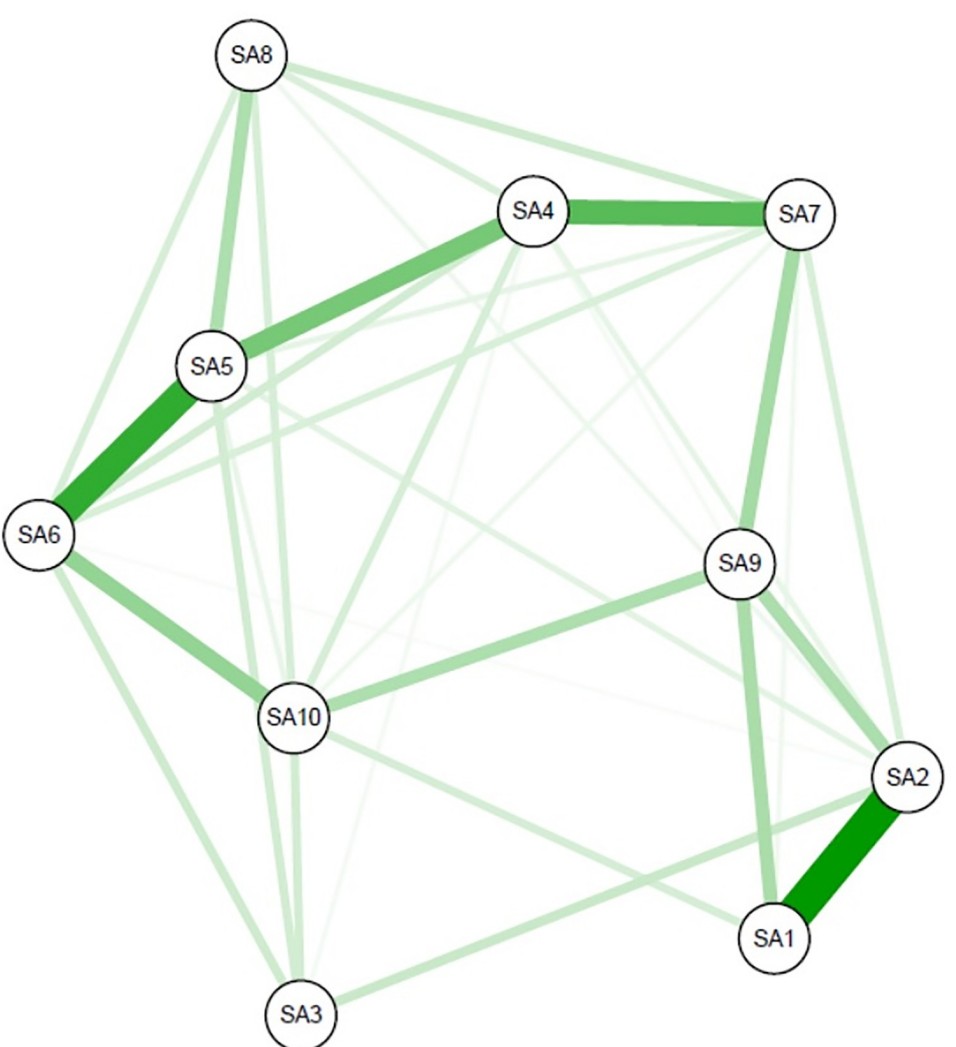

**Fig 1. Regularized partial correlation network of smartphone addiction scale short-version in university and college students in Japan (n = 478).** Green lines indicate a positive association. Line thickness reflects the strength of association, controlling for all other symptom nodes in the network. SA: Smartphone Addiction Scale–Short version item SA1: Missing planned work due to smartphone use, SA2: Having a hard time concentrating in class, while doing assignments, or while working due to smartphone use, SA3: Feeling pain in the wrists or at the back of the neck while using a smartphone, SA4: Won't be able to stand not having a smartphone, SA5: Feeling impatient and fretful when I am not holding my smartphone, SA6: Having my smartphone on my mind even when I am not using it, SA7: I will never give up using my smartphone even when my daily life is already greatly affected by it, SA8: Constantly checking my smartphone so as not to miss conversations between other people on Twitter or Facebook, SA9: Using my smartphone longer than I had intended, SA10: The people around me tell me that I use my smartphone too much.

In the given network, SA5 was considered a central node with the highest strength and high closeness (Fig 2). SA6 (Have my smartphone on my mind even when not using it) was with the highest closeness and relatively high strength in centrality indices. In particular, while this item was strongly associated with SA5, it also had the moderate association with SA10 (People around me tell me that I use my smartphone too much) that was only weakly associated with the central node (SA5). Both SA3 (Feel pain in the wrists or the neck while using a smartphone) and SA8 (Constantly check my smartphone not to miss information on SNS) were two items that notably had low centrality indices.

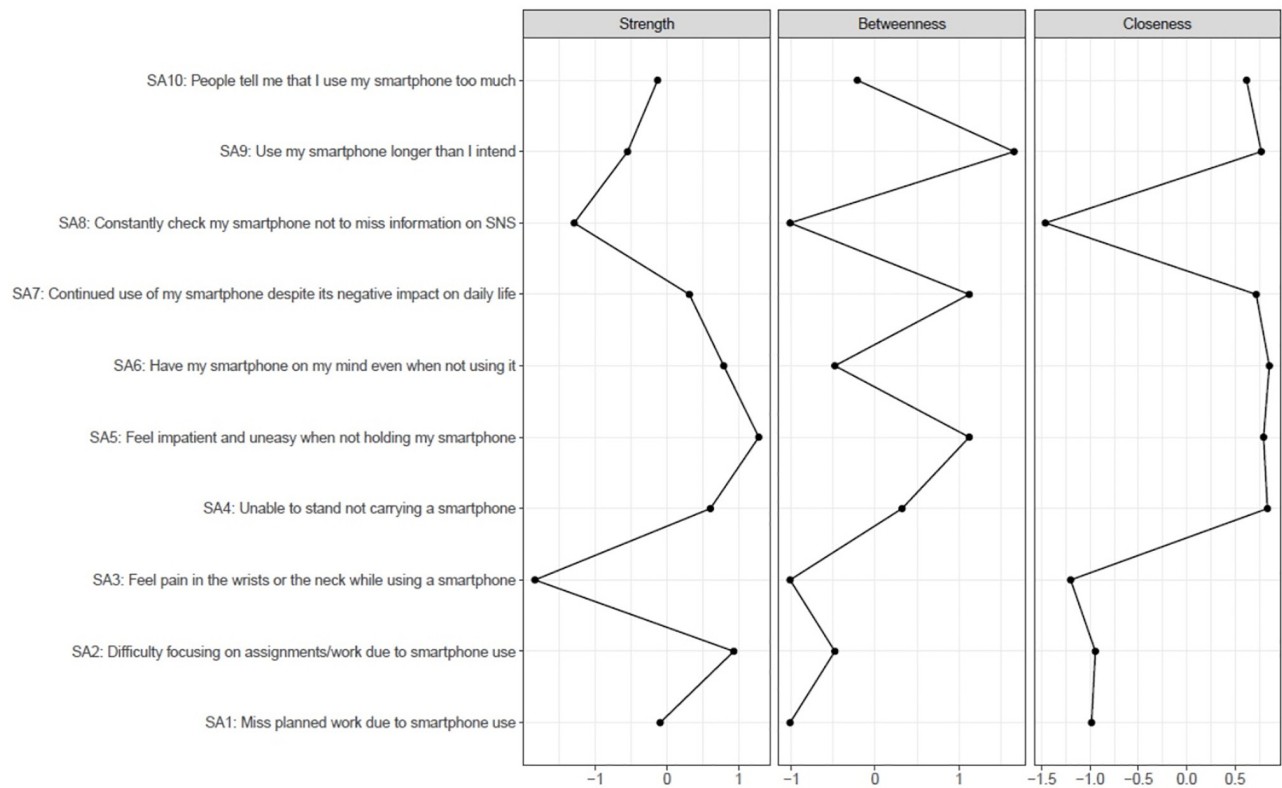

**Fig 2. Network centrality.** Values presented as standardized Z-scores, with higher scores indicative of greater influence within the overall network. SA: Smartphone Addiction Scale—Short version item.

### Network stability

The edge weights bootstrap (S2 File) showed that the 95% confidence intervals for many of the edges were overlapping. Additionally, there were few significant differences between the strong edges (S3 File). These findings indicate the most of the edges do not significantly differ and that the ranking of edge weights should be interpreted with care.

As shown in S4 File, examination of the stability of centrality indices was satisfactory for strength but not for other indices (the correlation stability coefficient of strength: 0.75, closeness: 0.44, and betweenness: 0.05).

### Discussion

In the present study, we estimated the network structure of problematic smartphone use symptoms among college students in Japan and identified the central (or "core") symptoms in the smartphone addiction psychopathology network. Although two recent studies examined the network structure of problematic smartphone use and smartphone addiction in school-age children in China [25] and Brazil [18], this is the first study that employed the network approach in understanding psychopathology of problematic smartphone use in college students in Japan who were transitioning to adulthood. This population has higher possession rates of smartphones and quite likely has scant or fewer parental controls compared to the school-age-sample previously studied [18, 25]. In fact, the survey conducted in Japan exhibited more maladaptive and excessive use of smartphones in college students compared to middle and high school students [36]. Thus, findings from the present study could further our

understanding of the network structure of problematic smartphone use psychopathology at this important developmental stage and could become the foundation for future research.

Our findings that identified one withdrawal symptom (SA5) and one preoccupation symptom (SA6) as central items in the network of problematic smartphone use were in line with those of the study conducted by Andrade et al. [18]. Consistent findings in the two studies targeting different age groups suggest these two symptoms play pivotal roles in the psychopathology of problematic smartphone use for a wide range of ages (from school-age children and adolescents to young adults). Furthermore, these central items and the strong edge (connection) between these two items in our study would support the three-stage model of addiction, where three stages of binge/intoxication (overuse of smartphone), withdrawal/negative affect, and preoccupation/anticipation, feed into each other to produce the addiction cycle [37]. Although we did not identify strong centrality in items pertaining to continued overuse of the smartphone (SA9 and SA10 items), it is still possible that these symptoms contribute to either the development or maintenance of the network of problematic smartphone use via the associations between these overuse symptoms and preoccupation symptoms (edge weight 0.21 between SA6 and SA10, for example) based on the above-mentioned three-stage model of addiction. It would be meaningful to examine the directionality among symptoms in order to better understand how the symptoms spread in the network of problematic smartphone use. However, doing so is beyond the scope of this study given our cross-sectional design. Future studies may be able to address this by using advanced data collection methods (intensive longitudinal data through the use of ecological momentary assessment [38], for example) that allow us to elucidate the directionality among symptoms and symptom dynamics in problematic smartphone use.

Huang and his colleagues conducted a network analysis of problematic smartphone use, measured by using the Smartphone Addiction Proneness Scale, a self-rating scale, in grade 4 and 8 students in China [25]. They reported that loss of control and continued excessive use were the central symptoms of problematic smartphone use. Differences in the symptom centrality between the study above and our study may be attributed to the difference in the scale used for each study as the functions of the scale modified by researchers in the above-mentioned study did not have withdrawal as one of the scale dimensions. Additionally, the difference could be due to the participants' age in the two studies given that the ability of self-control generally improves with age [39]. It is also reported that the older the age, the more people use their smartphones for social purposes, such as SNS [40]. When smartphone users communicate on SNS applications, message senders who do not receive a response sometimes perceive themselves as being neglected or ignored. Conversely, message recipients may become obsessive about checking and immediately replying to messages and keep their smartphones by their side during all their waking hours. Thus, such smartphone users could have difficulties leaving their smartphones out of reach and feel anxious without the smartphone in close proximity. Accordingly, we believe that young adult population, compared to school age population, might have withdrawal symptoms as more central in the network than overuse symptoms in the present study.

In the present study, more than half of the participants reported SNS as their primary use of the smartphone (62.2%), the percentage of which was higher than that of other reasons (gaming, for example). This raises a potential hypothesis that different smartphone usages lead to different network structures. Future studies with larger sample sizes would allow researchers to examine if the smartphone usage is an important factor contributing to network structures of problematic smartphone use.

This study has several limitations. The sample size was modest. Given that the stability of centrality indices in the network analysis is affected by the study sample size, larger sample-

sized studies are desirable. However, in the present study, the stability of strength, one of the centrality indices, was satisfactory. Regarding our sampling method, we recruited study participants only from colleges in one region in Japan, and the gender ratio was far from even, affecting the generalizability of our study findings. Additionally, our cross-sectional study design prohibited us from elucidating temporal relationships among individual symptoms of problematic smartphone use. As stated above, future studies may clarify the temporality of symptoms in problematic smartphone use by employing advanced data collection methods (intensive longitudinal data, for example) and analytic methods. Further investigations should be conducted to clarify the deeper mechanisms of problematic smartphone use in order to establish future effective prevention strategies, as well as possible treatment paradigms.

## Conclusions

Our results suggest that both withdrawal and preoccupation symptoms play pivotal roles in the psychopathology of problematic smartphone use in young adults. Longitudinal study designs are required to elucidate the role of these central items on the formation and maintenance of this behavioral problem.

## Supporting information

**S1 File. Edge weights in the smartphone addiction network in the study participants.** SAS-SV: Smartphone Addiction Scale–Short Version. (DOCX)

**S2 File. Bootstrapped 95% confidence intervals of edge weights.** The panel above presents edge weights of the estimated network of 10 Smartphone Addiction Scale items and 95% confidence intervals (CIs) calculated using bootstrapping. The red line represents the original sample values, the black dots represent the bootstrap means, and the gray areas indicate the bootstrapped CIs. Each horizontal line represents one edge of the network, ordered from the edge with the highest edge-weight to the edge with the lowest edge-weight. The y-axis labels were removed to avoid cluttering for the simplicity of the graph. Figure indicates that many edge-weights likely do not significantly differ from one-another. The generally large bootstrapped CIs imply that interpreting the order of most edges in the network should be done with care. (DOCX)

**S3 File. Edge difference test.** In this graph, each point on the x and y axes represents a pair of edges identified in a given network. Black boxes indicate significant differences between two edges (a bootstrap stability difference test: alpha = 0.05), whereas gray boxes do not indicate any significant differences. The diagonal represents the edge strength, where white indicates weak edge strengths, while blue indicates strong edge strengths. (DOCX)

**S4 File. Stability of network centrality.** The figure above presents the centrality stability as assessed using the case-dropping bootstrap method. Stability was assessed by re-estimating the network based on increasingly smaller subsets of the original sample. (DOCX)

**S5 File. R code used for data analysis in the present research.** (DOCX)

**S1 Data.** (XLSX)

## Author Contributions

**Conceptualization:** Masaru Tateno, Takahiro A. Kato, Tomoya Hirota.

**Data curation:** Masaru Tateno, Takahiro A. Kato, Tomohiro Shirasaka, Junichiro Kanazawa, Wataru Ukai.

**Formal analysis:** Tomoya Hirota.

**Funding acquisition:** Takahiro A. Kato.

**Investigation:** Masaru Tateno, Takahiro A. Kato.

**Methodology:** Tomoya Hirota.

**Project administration:** Masaru Tateno, Takahiro A. Kato, Tomoya Hirota.

**Software:** Tomoya Hirota.

**Supervision:** Takahiro A. Kato, Tomoya Hirota.

**Visualization:** Tomoya Hirota.

**Writing – original draft:** Masaru Tateno.

**Writing – review & editing:** Takahiro A. Kato, Tomohiro Shirasaka, Junichiro Kanazawa, Wataru Ukai, Tomoya Hirota.

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
