## [Decision Letter · Decision Letter 0]

5 Apr 2022

PONE-D-21-32886A network analysis of smartphone addiction symptoms in Japanese young adultsPLOS ONE

Dear Dr. Hirota,

Thank you for submitting your manuscript to PLOS ONE. After careful consideration, we feel that it has merit but does not fully meet PLOS ONE’s publication criteria as it currently stands. Therefore, we invite you to submit a revised version of the manuscript that addresses the points raised during the review process.

Specifically, I have now heard from two reviewers and both of them agree about the importance of the topic and study. The network analysis is soundly applied. Having said that, both reviewers came up with a number of comments (some major and some minor), which I hope you'd be able to address in a revision. Specifically, I hope that you would be able to strengthen the robustness of your discussion of the results.

We look forward to receiving your revised manuscript.

Kind regards,

Roland Bouffanais, Ph.D.

Academic Editor

PLOS ONE

https://journals.plos.org/plosone/s/file?id=ba62/PLOSOne_formatting_sample_title_authors_affiliations.pdf".

“This work was partially supported by the Grant-in-Aid for Scientific Research on (1) KAKENHI—the Japan Society for the Promotion of Science- JSPS (JP16H06403 & JP20H01773) and (2) The Japan Agency for Medical Research and Development - AMED (JP19dk0307073 & JP21wm0425010). The funders had no role in study design, data collection and analysis, in the decision to publish or the preparation of the manuscript.”

5. Thank you for stating the following in the Funding Section of your manuscript:

“This work was partially supported by the Grant-in-Aid for Scientific Research on (1) KAKENHI—the Japan Society for the Promotion of Science- JSPS (JP16H06403 & JP20H01773) and (2) The Japan Agency for Medical Research and Development - AMED (JP19dk0307073 & JP21wm0425010). The funders had no role in study design, data collection and analysis, in the decision to publish or the preparation of the manuscript.”

7. We note that you have indicated that data from this study are available upon request. PLOS only allows data to be available upon request if there are legal or ethical restrictions on sharing data publicly. For more information on unacceptable data access restrictions, please see http://journals.plos.org/plosone/s/data-availability#loc-unacceptable-data-access-restrictions.

8. Please note that in order to use the direct billing option the corresponding author must be affiliated with the chosen institute. Please either amend your manuscript to change the affiliation or corresponding author, or email us at plosone@plos.org with a request to remove this option.

9. Your ethics statement should only appear in the Methods section of your manuscript. If your ethics statement is written in any section besides the Methods, please delete it from any other section.

Additional Editor Comments:

I have now heard from two reviewers and both of them agree about the importance of the topic and study. The network analysis is soundly applied. Having said that, both reviewers came up with a number of comments (some major and some minor), which I hope you'd be able to address in a revision. Specifically, I hope that you would be able to strengthen the robustness of your discussion of the results.

Reviewers' comments:

Reviewer's Responses to Questions

**Comments to the Author**

1. Is the manuscript technically sound, and do the data support the conclusions?

Reviewer #1: Partly

Reviewer #2: Yes

2. Has the statistical analysis been performed appropriately and rigorously? 

Reviewer #1: Yes

Reviewer #2: Yes

3. Have the authors made all data underlying the findings in their manuscript fully available?

Reviewer #1: No

Reviewer #2: Yes

4. Is the manuscript presented in an intelligible fashion and written in standard English?

Reviewer #1: No

Reviewer #2: Yes

5. Review Comments to the Author

Reviewer #1: In this article the authors perform an exploratory network analysis of the answers to the smartphone addiction scale-short version of 487 Japanese college and university students. This included estimating the network and 3 centrality indices. The authors also proceed to divide the sample according to gender and estimate two sub-sample networks, testing for their differences. They found that preoccupation and withdrawal related items are particularly important in the network and also found differences between male and female subsample networks.

The topic of the article is of great importance and the use of network analysis could be of significance to further improve the knowledge about addiction disorders. The network analysis methodology is overall well applied. The findings about the most central symptoms could indeed be of interest to the field.

However, this reviewer thinks the study’s novelty is limited and also that it lacks a robust discussion of the results.

The overall quality of the writing is modest.

Major issues

- Study’s novelty: the cited study by Andrade, 2020 also performs a network analysis of the same instrument as the current study. More so, it frames it in a validation context of the instrument, making it a sounder scientific effort than the current one. The authors claim however that the fact that they’re using a slightly older sample justifies the study’s novelty. That difference seems to this reviewer as a very small one (in the end the results did not differ much, proving that the difference was not so relevant).

- Some questions about the use of this instrument (SAS-SV) should be clarified, in particular its validation and/or the validation of the original SAS instrument in Japan. Was that work done previously? In this reviewer’s perspective that would be a fundamental first step before engaging in a study like the present one (in the cited Brazilian study, a careful validation of the SAS-SV is performed alongside network analysis)

- Regarding the sample, it represents a group of students from a specific region of Japan and with high gender distribution imbalance. This limits generalizability as the authors point. However, a more detailed description of the sample and its recruitment is necessary. For instance: when was the data collected? In which schools, from which specific region? Was there a rational for choosing this sample? Could the authors explain better the setting in which the questionnaire was filled? Were there potential participants who did not answer to the questionnaire? Simultaneously, a better sociodemographic description of these individuals should be provided – detailed school level description, age range (min and max values), and ideally some information that could inform the reader about socioeconomic status of this sample.

- Regarding the study’s aims, the reviewer believes the network comparison performed is lacking a justification. The authors should explain why they believe it is relevant to compare networks based on a gender dichotomization of the sample. Also, one problem regarding this aim is the imbalance on sub sample size. Given that the starting point is already relatively modest, the authors are left with small samples for the sub-analysis, particularly the men’s network. Was the stability of each of the subsample networks analyzed?

- As a general comment, the discussion section should be improved. In the current form it stresses too much aspects that are limited by design in the current study (eg ideas about how addiction disorders are formed or should be treated).

- The writing should be improved. A native English speaker with experience in scientific writing could help improve substantially the article (see in minor issues some passages to reconsider).

- A justification for not providing the data should be given (PLoS Data policy); This reviewer also strongly encourages the authors to make data analysis information available as supplementary material (R code)

Minor issues

- Smartphone addiction concept: the authors point the lack of consensus around this construct. The reviewer would ask the authors to consider the use of ‘Problematic Smartphone Use’ instead

- On a more technical note about the network comparison performed, the authors determine that they will perform 3 different tests in their methods section. As van Borkulo et al point out in their paper explaining this procedure, the edge strength invariance test should only be performed if there is an a priori hypothesis about specific edges to directly test between the networks, or, in case there is no a priori hypothesis, it could be done as a follow-up to a significant difference in the structural invariance test. Therefore, in this reviewer’s opinion, since there is no a priori hypothesis for the authors, in the methods section it should be specified that this test (edge strength invariance) will only be performed if a difference in structural invariance is found. Then, in the results section, the network comparison tests report seems to be a bit unclear. The authors state that there was a “significant difference for the overall network structure” but explain this as meaning that the “nodes were more densely connected overall in female students than in male students”. This explanation refers to the invariant global strength aspect of the test and not the invariant network structure. So, this reviewer is confused about what is being reported – invariant network structure or invariant global strength? The edge weight comparisons reported afterwards should only be pursued in case network structure differs. (See: Van Borkulo, C. D., Boschloo, L., Kossakowski, J., Tio, P., Schoevers, R. A., Borsboom, D., & Waldorp, L. J. (2017). Comparing network structures on three aspects: A permutation test. )

- The meaning of centrality indices should be rectified. On page 13 line 4, the way strength centrality is presented is not correct (ie, strength centrality does not translate the association of a node with “all other other nodes in the network” as stated. It is instead a measure of local influence). The meaning of betweenness and closeness centrality measures on cross-sectional data network analysis in the context of mental disorders has also been questioned. I would suggest the author to check references such as the following for this matter: Bringmann, L. F., Elmer, T., Epskamp, S., Krause, R. W., Schoch, D., Wichers, M., ... & Snippe, E. (2019). What do centrality measures measure in psychological networks?. Journal of abnormal psychology, 128(8), 892.

- In the methods section it should be mentioned which option was chosen for the network layout.

- Table 1: the asterisks in the Age and SAS-SV lines are a bit confusing and seem unnecessary to this reviewer; the readability of the table could be improved if the “(mean±SD” parts were positioned without a line break and if some hierarchical visual cue could be added to subdivisions of “internet use (hrs) and “purpose”; the acronym SNS should be explained in the table legend.

- Network estimation subsection from analytic plans: please use proper reference for qgraph package. A reference for the R statistical software is also lacking.

- Network estimation subsection from analytic plans: the sentence “leading to a sparse network with explanatory power”. What is meant by 'explanatory power'?

- Results, network structure and centrality, paragraph2: “The weights of the edges between the items SA1 (…) and SA2 (…) and the items SA5 (…) were stronger than others” – it is unclear which 2 pairs of nodes are the authors referring too. Also, the use of “stronger than others” in this context seems imprecise as this is not being tested.

- Results, network structure and centrality, paragraph3: “Inspecting the network structure” – the observations in this sentence do not follow from “inspection” of the network structure but instead of the centrality values. The visual layout of the network is rather arbitrary.

- Results, network structure and centrality, paragraph3: “strongly associated with SA5, the central node” please rephrase. The qualification “the central node” is imprecise. SA5 is the node with highest strength centrality.

- Results, network structure and centrality, paragraph3: “moderate association with SA9”. It seems that SA10 is meant here instead of SA9.

- Discussion paragraph 1: "findings from the present study could further our understanding of how smartphone addiction psychopathology is developed and maintained" - please reconsider. Cross-sectional network analysis cannot inform about dynamic aspects of mental disorders such as their development or maintenance.

- Discussion, paragraph 4: please consider rephrasing how you draw conclusions from your study to therapeutic approaches. It seems that the current formulation is too strong given the limitations of the current study design and implementation.

- Discussion, paragraph 4: There is a long description of current approaches to treat behavioral addictions. This is however disconnected from the current study’s results (and purposes) and therefore seems misplaced in the discussion.

- Discussion, paragraph 5: The explanation given about the differences between male and female subsample networks is not very sound in the opinion of this reviewer. It is unclear how “different psychological backgrounds” (a concept that is not very specific) would lead to “strong edge weights”. The remark about a supposed female ‘fear of missing out’ seems arbitrarily used. It should be discussed however if the differences in connectivity found could be attributed in the first place to female higher scores and a ‘floor-effect’ generated by lower scores on the male side.

- Conclusions: the use of “formation” of psychopathology is inaccurate because a cross-sectional network cannot inform about causality or dynamic patterns related to how psychopathology emerges.

- Conclusions: the sentence “Findings from this network analysis would provide us with deeper understandings of smartphone addiction” is too vague. In the end the reader struggles to find this study's take-away messages.

- English writing aspects – the reviewer suggests that the text is reviewed by a native English speaker. Here are however some suggestions from the reviewer:

-Introduction, second paragraph: the reviewer would suggest the authors to remove the sentence “The superb mobility and multifunction capability allows us to access the internet anytime and anywhere.

-Introduction, paragraph 3: “smartphone addiction is considered [A] behavioral addiction (A is lacking); “which a person is forced to engage – the use of “forced” seems too strong, please use a more adequate alternative;

-Introduction, paragraph 5: “including THE smartphone addiction scale” (THE is lacking)

-Introduction, paragraph 6: “behavioral addiction is complex human behaviors” should become “behavioral addiction is A complex human BEHAVIOR”

-Introduction, paragraph 7: “Rather, it is considered a ring as a complex network of mutually reinforcing symptoms” – this sentence is not clear, please rephrase, and consider changing the word “ring”.

-The section termed “analytic plans” should in this reviewer's view be renamed. Consider alternatives like “data analyses” or use “network analysis” as an umbrella that you then subdivide into subanalyses.

-Network estimation subsection: “represent individual signs (symptoms)”, consider using the term "item” as in this case the nodes represent items from an instrument.

-Centrality subsection: “sum of distances from the node to all other nodes):” use ";" instead of ":" here

-Centrality subsection: “we interpreted a symptom with the strongest strength as a central symptom in a given network” please rephrase

-Discussion, paragraph 1: “identified the important symptoms “. Rephrase paying attention to the fact that the use of THE important symptoms is inaccurate.

-Discussion, paragraph 3: “Huang and his colleagues conducted a network analysis of problematic smartphone use in grade 4 and 8 students in China using a self-rating scale, smartphone addiction proneness scale” – rewrite the end of the sentence

-“central indices” should be corrected to “centrality indices”

Reviewer #2: It is my pleasure to read this study with adequate sample and exact analysis. I have some suggestion for improving the manuscript.

1 Please include the theory or evidence for the addiction characteristics of excessive smart phone use in the introduction section.

2 The female sample is larger than male. Please explain the reason and detail for possible limitation, especially for the gender effect on smartphone use.

3 Since there is only ten item in the scale, please discuss whether the number of item will limit the analysis for the items.

4 Pleas explain the result of the figure 1 in the revsied manuscript

5 Although s4 and s5 are the central of the network, however, the score in these two item was relative lower. Please explain it.

6 The author had well demonstraed the implication of the result of the study. I agree with most of their claim. However, this content was not proved in this study. Thus, it should prevent to provide this content as it had been proved. Other reference should be provide to support their claim.

6. PLOS authors have the option to publish the peer review history of their article (what does this mean?). If published, this will include your full peer review and any attached files.

Reviewer #1: No

Reviewer #2: No

---

## [Author Response · Author response to Decision Letter 0]

1 Jun 2022

Our replies to reviewers’ comments

Reviewer #1: In this article the authors perform an exploratory network analysis of the answers to the smartphone addiction scale-short version of 487 Japanese college and university students. This included estimating the network and 3 centrality indices. The authors also proceed to divide the sample according to gender and estimate two sub-sample networks, testing for their differences. They found that preoccupation and withdrawal related items are particularly important in the network and also found differences between male and female subsample networks.

The topic of the article is of great importance and the use of network analysis could be of significance to further improve the knowledge about addiction disorders. The network analysis methodology is overall well applied. The findings about the most central symptoms could indeed be of interest to the field.

However, this reviewer thinks the study’s novelty is limited and also that it lacks a robust discussion of the results.

The overall quality of the writing is modest.

Major issues

- Study’s novelty: the cited study by Andrade, 2020 also performs a network analysis of the same instrument as the current study. More so, it frames it in a validation context of the instrument, making it a sounder scientific effort than the current one. The authors claim however that the fact that they’re using a slightly older sample justifies the study’s novelty. That difference seems to this reviewer as a very small one (in the end the results did not differ much, proving that the difference was not so relevant).

Our reply: Thank you for your comment and input. We acknowledge that the fact we focused on different age samples (transitional age youth) from ones in the extant study may not sound substantially novel. However, we would like to emphasize that understanding how this addictive behavior functions at different developmental age is important as the ownership of smartphones and degree of parental control and possibly school rules (for students who attend compulsory school, such as middle school and high school) can influence problematic smartphone behavior. We added a sentence and a reference below to state that maladaptive and excessive use of smartphones was identified more frequently in college students than middle and high school students in the revised manuscript to support our concept. 

Discussion 1st paragraph lines 11 – 13: “In fact, the survey conducted in Japan exhibited more maladaptive and excessive use of smartphones in college students compared to middle and high school students [35]”

- Some questions about the use of this instrument (SAS-SV) should be clarified, in particular its validation and/or the validation of the original SAS instrument in Japan. Was that work done previously? In this reviewer’s perspective that would be a fundamental first step before engaging in a study like the present one (in the cited Brazilian study, a careful validation of the SAS-SV is performed alongside network analysis)

Our reply: The original instrument (SAS) was validated in Japanese youths in another study, where the authors reported the Japanese-version of SAS demonstrated good reliability and validity (DOI: 10.1007/s11469-021-00594-z). We added this statement in the revised manuscript. We acknowledge that the short-version (SAS-SV) has not validated in Japanese samples yet.

Lines 5-6 in Measures > Smartphone Addiction Scale-Short-version: “The Japanese-version of the SAS was developed, and its reliability and factor validity were assessed in 1,037 youth aged 15 – 24 years [28].”

- Regarding the sample, it represents a group of students from a specific region of Japan and with high gender distribution imbalance. This limits generalizability as the authors point. However, a more detailed description of the sample and its recruitment is necessary. For instance: when was the data collected? In which schools, from which specific region? Was there a rational for choosing this sample? Could the authors explain better the setting in which the questionnaire was filled? Were there potential participants who did not answer to the questionnaire? Simultaneously, a better sociodemographic description of these individuals should be provided – detailed school level description, age range (min and max values), and ideally some information that could inform the reader about socioeconomic status of this sample.

Our reply: We added descriptions of the study and study participants in the revised manuscript. 

Participants subsection in the Material and methods section: “The subjects of the study were 487 private college and university students in Sapporo city and its environs in Japan. Study participation rate was 81.2% (487/600). Their academic deviation scores were average or a little below the average compared to the national standard in Japan. Research collaborators for data collection were recruited through personal connections of the first author of this paper (MT). Nine teachers from three universities and six colleges agreed to voluntarily support our investigation.”

We did not collect socioeconomic status or detailed characteristic information about participants except for that already reported in the manuscript. 

- Regarding the study’s aims, the reviewer believes the network comparison performed is lacking a justification. The authors should explain why they believe it is relevant to compare networks based on a gender dichotomization of the sample. Also, one problem regarding this aim is the imbalance on sub sample size. Given that the starting point is already relatively modest, the authors are left with small samples for the sub-analysis, particularly the men’s network. Was the stability of each of the subsample networks analyzed?

Our reply: We agree with the reviewer’s point. We acknowledge that the comparison test was exploratory without sufficient scientific justification. Additionally, as pointed by the reviewer, even we conduct the network comparison test, the sample size differences between female and male students and the small sub-sample size of male students would result in skewed findings. Thus, the authors determined to omit this analysis and remove corresponding paragraphs from the manuscript. 

- As a general comment, the discussion section should be improved. In the current form it stresses too much aspects that are limited by design in the current study (eg ideas about how addiction disorders are formed or should be treated).

Our reply: Thank you so much for your comments. We revised our manuscript, including discussion based on reviewers’ feedback. 

- The writing should be improved. A native English speaker with experience in scientific writing could help improve substantially the article (see in minor issues some passages to reconsider).

Our reply: We asked a native English speaker who holds a doctor’s degree and is an acquaintance of one of the authors (MT) to proofread the manuscript. 

- A justification for not providing the data should be given (PLoS Data policy); This reviewer also strongly encourages the authors to make data analysis information available as supplementary material (R code)

Our reply: We uploaded the R code use for the analysis in this study in Supplementary data 5. 

Minor issues

- Smartphone addiction concept: the authors point the lack of consensus around this construct. The reviewer would ask the authors to consider the use of ‘Problematic Smartphone Use’ instead

Our reply: We agree with the authors’ suggestion. We changed the term “smartphone addiction” to “problematic smartphone” throughout the manuscript.

- On a more technical note about the network comparison performed, the authors determine that they will perform 3 different tests in their methods section. As van Borkulo et al point out in their paper explaining this procedure, the edge strength invariance test should only be performed if there is an a priori hypothesis about specific edges to directly test between the networks, or, in case there is no a priori hypothesis, it could be done as a follow-up to a significant difference in the structural invariance test. Therefore, in this reviewer’s opinion, since there is no a priori hypothesis for the authors, in the methods section it should be specified that this test (edge strength invariance) will only be performed if a difference in structural invariance is found. Then, in the results section, the network comparison tests report seems to be a bit unclear. The authors state that there was a “significant difference for the overall network structure” but explain this as meaning that the “nodes were more densely connected overall in female students than in male students”. This explanation refers to the invariant global strength aspect of the test and not the invariant network structure. So, this reviewer is confused about what is being reported – invariant network structure or invariant global strength? The edge weight comparisons reported afterwards should only be pursued in case network structure differs. (See: Van Borkulo, C. D., Boschloo, L., Kossakowski, J., Tio, P., Schoevers, R. A., Borsboom, D., & Waldorp, L. J. (2017). Comparing network structures on three aspects: A permutation test. )

Our reply: Thank you for suggested papers and your constructive comments. We agree with the reviewer’s points and concerns. In the revised manuscript, we determined not to conduct the network comparison test given no priori hypotheses driving this analysis and as the study findings from this analysis could be skewed and may lack the accuracy due to the small sub-sample size of male students and inequality of sample size between female and male students. 

- The meaning of centrality indices should be rectified. On page 13 line 4, the way strength centrality is presented is not correct (ie, strength centrality does not translate the association of a node with “all other other nodes in the network” as stated. It is instead a measure of local influence). The meaning of betweenness and closeness centrality measures on cross-sectional data network analysis in the context of mental disorders has also been questioned. I would suggest the author to check references such as the following for this matter: Bringmann, L. F., Elmer, T., Epskamp, S., Krause, R. W., Schoch, D., Wichers, M., ... & Snippe, E. (2019). What do centrality measures measure in psychological networks?. Journal of abnormal psychology, 128(8), 892.

Our reply: The authors very much appreciate the reviewer’s feedback and suggested papers. We addressed this point in the revised manuscript (see the “Centrality” subsection in the “Material and methods” section). 

- In the methods section it should be mentioned which option was chosen for the network layout.

Our reply: We used the Fruchterman-Reingold algorithm, which we added to the revised manuscript (the last sentence in the network estimation subsection under the Data analyses section).

- Table 1: the asterisks in the Age and SAS-SV lines are a bit confusing and seem unnecessary to this reviewer; the readability of the table could be improved if the “(mean±SD” parts were positioned without a line break and if some hierarchical visual cue could be added to subdivisions of “internet use (hrs) and “purpose”; the acronym SNS should be explained in the table legend.

Our reply: All addressed in the revised manuscript. 

- Network estimation subsection from analytic plans: please use proper reference for qgraph package. A reference for the R statistical software is also lacking.

Our reply: The reference is now placed in the revised manuscript (#30). Thank you. 

- Network estimation subsection from analytic plans: the sentence “leading to a sparse network with explanatory power”. What is meant by 'explanatory power'?

Our reply: We decided not to add the phrase pointed by the reviewer as the statement was vague and confusing. 

- Results, network structure and centrality, paragraph2: “The weights of the edges between the items SA1 (…) and SA2 (…) and the items SA5 (…) were stronger than others” – it is unclear which 2 pairs of nodes are the authors referring too. Also, the use of “stronger than others” in this context seems imprecise as this is not being tested.

Our reply: Thank you for pointing this out. To clarify our intents, we revised the manuscript as follows (changes are highlighted in yellow): “The weights of the edges between the items SA1 (Miss planned work due to smartphone use) and SA2 (Difficulty focusing on assignments/work due to smartphone use) and those between the items SA5 (Feel impatient and uneasy when not holding my smartphone) were larger than others in the given network (0.49 and 0.41, respectively).”

- Results, network structure and centrality, paragraph3: “Inspecting the network structure” – the observations in this sentence do not follow from “inspection” of the network structure but instead of the centrality values. The visual layout of the network is rather arbitrary.

Our reply: We agree with the reviewer’s point. We deleted that phrase in the revised manuscript. Corresponding sentence in the revised manuscript is below: “SA6 (Have my smartphone on my mind even when not using it) was with the highest closeness and relatively high strength in centrality indices.”

- Results, network structure and centrality, paragraph3: “strongly associated with SA5, the central node” please rephrase. The qualification “the central node” is imprecise. SA5 is the node with highest strength centrality.

Our reply: Thank you for this feedback. We corrected the corresponding sentence as follows in the revised manuscript. 

“In particular, while this item was strongly associated with SA5, it also had the moderate association with SA10 (People around me tell me that I use my smartphone too much) that was only weakly associated with the node with the highest strength centrality (SA5).”

- Results, network structure and centrality, paragraph3: “moderate association with SA9”. It seems that SA10 is meant here instead of SA9.

Our reply: It was a typo. The reviewer was right. We corrected this error in the revised manuscript. 

- Discussion paragraph 1: "findings from the present study could further our understanding of how smartphone addiction psychopathology is developed and maintained" - please reconsider. Cross-sectional network analysis cannot inform about dynamic aspects of mental disorders such as their development or maintenance.

- Discussion, paragraph 4: please consider rephrasing how you draw conclusions from your study to therapeutic approaches. It seems that the current formulation is too strong given the limitations of the current study design and implementation.

- Discussion, paragraph 4: There is a long description of current approaches to treat behavioral addictions. This is however disconnected from the current study’s results (and purposes) and therefore seems misplaced in the discussion.

- Discussion, paragraph 5: The explanation given about the differences between male and female subsample networks is not very sound in the opinion of this reviewer. It is unclear how “different psychological backgrounds” (a concept that is not very specific) would lead to “strong edge weights”. The remark about a supposed female ‘fear of missing out’ seems arbitrarily used. It should be discussed however if the differences in connectivity found could be attributed in the first place to female higher scores and a ‘floor-effect’ generated by lower scores on the male side.

- Conclusions: the use of “formation” of psychopathology is inaccurate because a cross-sectional network cannot inform about causality or dynamic patterns related to how psychopathology emerges.

- Conclusions: the sentence “Findings from this network analysis would provide us with deeper understandings of smartphone addiction” is too vague. In the end the reader struggles to find this study's take-away messages.

- English writing aspects – the reviewer suggests that the text is reviewed by a native English speaker. Here are however some suggestions from the reviewer:

-Introduction, second paragraph: the reviewer would suggest the authors to remove the sentence “The superb mobility and multifunction capability allows us to access the internet anytime and anywhere.

-Introduction, paragraph 3: “smartphone addiction is considered [A] behavioral addiction (A is lacking); “which a person is forced to engage – the use of “forced” seems too strong, please use a more adequate alternative;

-Introduction, paragraph 5: “including THE smartphone addiction scale” (THE is lacking)

-Introduction, paragraph 6: “behavioral addiction is complex human behaviors” should become “behavioral addiction is A complex human BEHAVIOR”

-Introduction, paragraph 7: “Rather, it is considered a ring as a complex network of mutually reinforcing symptoms” – this sentence is not clear, please rephrase, and consider changing the word “ring”.

-The section termed “analytic plans” should in this reviewer's view be renamed. Consider alternatives like “data analyses” or use “network analysis” as an umbrella that you then subdivide into subanalyses.

-Network estimation subsection: “represent individual signs (symptoms)”, consider using the term "item” as in this case the nodes represent items from an instrument.

-Centrality subsection: “sum of distances from the node to all other nodes):” use ";" instead of ":" here

-Centrality subsection: “we interpreted a symptom with the strongest strength as a central symptom in a given network” please rephrase

-Discussion, paragraph 1: “identified the important symptoms “. Rephrase paying attention to the fact that the use of THE important symptoms is inaccurate.

-Discussion, paragraph 3: “Huang and his colleagues conducted a network analysis of problematic smartphone use in grade 4 and 8 students in China using a self-rating scale, smartphone addiction proneness scale” – rewrite the end of the sentence

-“central indices” should be corrected to “centrality indices”

Our reply: Thank you so much for these suggestions. We revised our manuscript based on these suggestions (corrected parts are highlighted in yellow in the revised manuscript). In addition, as stated above in our letter, we asked a native English speaker to proofread the revised manuscript. 

Reviewer #2: It is my pleasure to read this study with adequate sample and exact analysis. I have some suggestion for improving the manuscript.

1 Please include the theory or evidence for the addiction characteristics of excessive smart phone use in the introduction section.

Our reply: As pointed out by the other reviewer, due to the lack of consensus on the constructs of smartphone addiction and no compelling theory supporting the concept of smartphone addiction at this point, we decided to use “problematic smartphone use” instead throughout the revised manuscript.

2 The female sample is larger than male. Please explain the reason and detail for possible limitation, especially for the gender effect on smartphone use.

Our reply: As answered above to respond to the other reviewer, we determined not to conduct the network comparison test as the study findings from this analysis could be skewed and may lack the accuracy due to the small sub-sample size of male students and inequality of sample size between female and male students. 

3 Since there is only ten item in the scale, please discuss whether the number of item will limit the analysis for the items.

Our reply: The findings from network analysis depend on what items are entered in analysis; however, this does not mean the number of items in the scale influence accuracy and stability of network analysis findings. 

4 Pleas explain the result of the figure 1 in the revsied manuscript

Our reply: Thank you for this comment. Figure 1 is only for visualization of network analytic findings. As discussed in other network analysis literature, it is recommended we discuss findings obtained from centrality analyses (figure 2) as visualization per se may be arbitrary. General description about network visualization is placed as a figure legend “Figure 1. Regularized partial correlation network of Smartphone Addiction Scale Short-Version in college and university students in Japan (N = 478). Green lines indicate a positive association. Line thickness reflects the strength of association, controlling for all other symptom nodes in the network.” 

5 Although s4 and s5 are the central of the network, however, the score in these two item was relative lower. Please explain it.

Our reply: Strength centrality is a measure that reflects the degree to which each node is connected to other nodes in the network. Thus, the degree of centrality differs from the degree of item score. 

6 The author had well demonstraed the implication of the result of the study. I agree with most of their claim. However, this content was not proved in this study. Thus, it should prevent to provide this content as it had been proved. Other reference should be provide to support their claim.

Our reply: Thank you for this comment. We agree with the reviewer’s point that the paragraph pertaining to the implication of the result of the study was not stated based on scientifically sound hypotheses. To avoid any confusion and to make the flow of the manuscript more coherent, we deleted this paragraph in the revised manuscript.

---

## [Decision Letter · Decision Letter 1]

30 Jun 2022

PONE-D-21-32886R1A Network Analysis of Problematic Smartphone Use in Japanese young adultsPLOS ONE

Dear Dr. Hirota,

Thank you for submitting your manuscript to PLOS ONE. After careful consideration, we feel that it has merit but does not fully meet PLOS ONE’s publication criteria as it currently stands. Therefore, we invite you to submit a revised version of the manuscript that addresses the points raised during the review process.

We have now received reports from the same two reviewers, and after careful consideration, we have decided to invite yet another major revision of the manuscript.

We look forward to receiving your revised manuscript.

Kind regards,

Roland Bouffanais, Ph.D.

Academic Editor

PLOS ONE

Additional Editor Comments (if provided):

We have now received reports from the same two reviewers, and after careful consideration, we have decided to invite another major revision of the manuscript.

As you will see from the enclosed reports, one reviewer still raises important concerns. We find that these concerns limit the strength of the study, and therefore we ask you to address them with additional work.

If you feel that you are able to comprehensively address the reviewer’s concerns, please provide a point-by-point response to these comments along with your revision. Please show all changes in the manuscript text file with track changes or color highlighting. If you are unable to address specific reviewer requests or find any points invalid, please explain why in the point-by-point response.

Reviewers' comments:

Reviewer's Responses to Questions

**Comments to the Author**

1. If the authors have adequately addressed your comments raised in a previous round of review and you feel that this manuscript is now acceptable for publication, you may indicate that here to bypass the “Comments to the Author” section, enter your conflict of interest statement in the “Confidential to Editor” section, and submit your "Accept" recommendation.

Reviewer #1: (No Response)

Reviewer #2: All comments have been addressed

2. Is the manuscript technically sound, and do the data support the conclusions?

Reviewer #1: Yes

Reviewer #2: Yes

3. Has the statistical analysis been performed appropriately and rigorously? 

Reviewer #1: Yes

Reviewer #2: Yes

4. Have the authors made all data underlying the findings in their manuscript fully available?

Reviewer #1: Yes

Reviewer #2: (No Response)

5. Is the manuscript presented in an intelligible fashion and written in standard English?

Reviewer #1: Yes

Reviewer #2: Yes

6. Review Comments to the Author

Reviewer #1: The authors conducted several changes in the manuscript that contributed to a much stronger work. Namely, the writing quality is now much improved, the analyses performed and their justification became clearer, and the availability of data and code now made this a transparent and potentially replicable scientific work.

There are some aspects though that I believe are still in need of revision in order for the manuscript to be acceptable for publication.

One general point is that, now that some unnecessary remarks were removed from the discussion, this section should be improved by adding thoughts about some of the issues the manuscript raises (see suggestions below).

(I acknowledge that some of my current minor points address issues already present in the first draft of the manuscript. Given the very high amount of changes required for the first revision, these points seemed less relevant then and/or are now in more clear need of revision given the other changes made to the article.)

Major issues

- In the abstract it is still mentioned that gender differences were explored in the network. This should be removed as that analysis is no longer performed.

- Table 1 has separate columns for male and female participants. Given that there is no longer a network comparison according to gender, could the authors state why this is still relevant? I would also like to know the authors’ opinion about 2 further aspects related to this, that might be relevant for the discussion: 1 ) why is it that ¾ of the sample are females (Is this something that mirrors the high-school/university demographics from Japan or is it a selection bias?) and how could that imbalance bias the results? 2) how could the different smartphone usages (reported in table 1) affect the results (e.g. can we consider this study as mostly representing social network use disorder? Could different usages lead to different network structures hypothetically? Would this factor be more important than gender itself?). I believe these aspects are very important to be considered in the discussion and would enrich it very much.

Minor issues

- Abstract (and results too): I believe it is confusing to state “The estimated network yielded 45 edges, among which 34 edges had non- zero weights”. What are the 0 weight edges? Maybe just state that you’ve found 34 edges.

- Intro (last sentence of the 2nd to last paragraph) “population would not arguably be influenced by family”. I suggest that you consider stating something like “would not be directly influenced by family”, given that how a family raised the subject would continue to influence him/her afterwards.

- Intro (last paragraph): Second aim “identify which items could play important roles” – to play an important role is already an extrapolation. What we can do is to identify the most central symptoms in the network (hence, as you say before, the symptoms that potentially play a critical role in the onset and maintenance of the disorder). Consider rephrasing.

- Table 1 – consider reducing the decimal places to 3, but check exactly the journal guidelines on this. P-values less than 0.001 are usually reported as p<0.001.

- Is there some information available about the nearly 200 students who did not answer the questionnaire? For instance, about gender (were they predominantly male? this could explain your sample bias).

- Methods: citation for the qgraph package should be added (in data analyses - network estimation) https://cran.r-project.org/web/packages/qgraph/citation.html#:~:text=qgraph%20citation%20info,4)%2C%201%E2%80%9318.

- Methods: please consider adding succinctly what the Fruchterman-Reingold algorithm does in terms of network layout (readers not acquainted with network methods might find it intriguing)

- Data analyses->centrality (first sentence): “We calculated several indices” – please consider rephrasing to “calculated three indices”

- Data analyses->centrality (second sentence): “a symptom with the highest value of strength was defined as a central symptom in a given network”. Consider rephrasing or removing this sentence as it is self-evident that the centrality is higher according to how high strength centrality is.

- Data analyses->centrality (last sentence): “These are typically presented as standardized Z-scores”. Consider clarifying – “these” -> centrality values

- On the first subparagraph of the results section, the text repeats what can be already found in table 1. I would strongly recommend that the authors simply refer to the table for the information that can already be found there.

- Results -> network structure and stability (first sentence): “relations among SA symptoms”. I believe you dropped the SA terminology, so please consider changing it consistently in the manuscript. This includes changing the node names in the figures. For example, instead of using SA#, consider simply using the number 1-10 (as you do in table 2).

- Results -> network structure and stability (second paragraph, first sentence): “The weights of the edges between the items SA1 (…) were larger than others in the given network“. Please consider rephrasing as this does not seem to have been specifically tested (and you mentioned there is considerable overlap in confidence intervals).

- Discussion: the term ‘smartphone addiction’ is sometimes used here in a way that is not completely harmonized within the manuscript given that there was an intention to substitute this term.

- Discussion (second paragraph, first sentence): “the Brazilian study” – please consider changing this. Would prefer either that you cite the authors name or that you refer to the “study conducted with [demographic group] in Brazil”

- Discussion (second paragraph, third sentence): “to produce [the] addiction cycle”

- Discussion (second paragraph, fourth sentence): consider enriching the discussion by explaining how you believe overuse symptoms could still “contribute to either the development or maintenance of the network” even though your results do not seem to show them as central.

- Discussion (third paragraph, third sentence): “Differences in the symptom centrality between the study above and our study may be attributed to the difference in the scale used for each study.” Please consider enriching your discussion by stating your thoughts on why that is the case. What are the differences between the instruments?

- Discussion (third paragraph, last sentence): “Thus, such smartphone users could have difficulties leaving their smartphones out of reach and feel anxious without the smartphone in close proximity.”. Good point. Consider making explicit that this is why you believe that an older population (like the one in your study) might have withdrawal symptoms as more central in the network than overuse symptoms.

- Figure 2 legend: “with higher scores indicative of greater importance within the overall network.” Same as stated above. Consider refraining from using the term “importance” as this is an extrapolation.

Reviewer #2: I appreciate the response of the authors. The manuscript had been well revised. It is fine to be published.

7. PLOS authors have the option to publish the peer review history of their article (what does this mean?). If published, this will include your full peer review and any attached files.

Reviewer #1: **Yes: **Bernardo Melo Moura

Reviewer #2: No

---

## [Author Response · Author response to Decision Letter 1]

17 Jul 2022

Our responses to the reviewer's comments are summarized in the document and uploaded as a separate file (entitled as "Response to the reviewer_PLOSONE_R2"). Thank you.

---

## [Decision Letter · Decision Letter 2]

27 Jul 2022

A Network Analysis of Problematic Smartphone Use in Japanese young adults

PONE-D-21-32886R2

Dear Dr. Hirota,

We’re pleased to inform you that your manuscript has been judged scientifically suitable for publication and will be formally accepted for publication once it meets all outstanding technical requirements.

Kind regards,

Roland Bouffanais, Ph.D.

Academic Editor

PLOS ONE

Additional Editor Comments (optional):

Reviewers' comments:

Reviewer's Responses to Questions

**Comments to the Author**

1. If the authors have adequately addressed your comments raised in a previous round of review and you feel that this manuscript is now acceptable for publication, you may indicate that here to bypass the “Comments to the Author” section, enter your conflict of interest statement in the “Confidential to Editor” section, and submit your "Accept" recommendation.

Reviewer #1: All comments have been addressed

Reviewer #2: All comments have been addressed

2. Is the manuscript technically sound, and do the data support the conclusions?

Reviewer #1: Yes

Reviewer #2: Yes

3. Has the statistical analysis been performed appropriately and rigorously? 

Reviewer #1: Yes

Reviewer #2: Yes

4. Have the authors made all data underlying the findings in their manuscript fully available?

Reviewer #1: Yes

Reviewer #2: Yes

5. Is the manuscript presented in an intelligible fashion and written in standard English?

Reviewer #1: Yes

Reviewer #2: Yes

6. Review Comments to the Author

Reviewer #1: The authors' revision of the manuscript contributed again to improve substantially their work.

There are 4-5 minor points I will still list below for the authors to revise. Granted that attention is paid to these points, I do not believe a further review is necessary, thus the paper is in my opinion good enough for publication.

Minor points

- Pay attention to the fact that the abstract you provide in the box section at the beginning does not exactly match the one present on the manuscript file you’ve attached

- Abstract, Results: I would suggest the following change in writing: “We identified 34 edges in the estimated network. One item pertaining to withdrawal symptoms had the highest strength and also high closeness centrality.

- Results, characteristics of the study participants: also for male students add years of age “(…) male students (mean years of age: …”

- Results, network structure and stability: Use of “SA”. Consider using “SAS-SV items” instead of “SA symptoms”

- Results, network structure and stability: I did not spot this before, but it is important to correct: “and those between the items SA5 (…)” it seems like the other node is missing, please add mention to it.

Reviewer #2: (No Response)

7. PLOS authors have the option to publish the peer review history of their article (what does this mean?). If published, this will include your full peer review and any attached files.

Reviewer #1: **Yes: **Bernardo Melo Moura

Reviewer #2: No

---

## [Editor Report · Acceptance letter]

29 Jul 2022

PONE-D-21-32886R2 

A Network Analysis of Problematic Smartphone Use in Japanese young adults 

Dear Dr. Hirota:

I'm pleased to inform you that your manuscript has been deemed suitable for publication in PLOS ONE. Congratulations! Your manuscript is now with our production department. 

Kind regards, 

on behalf of

Professor Roland Bouffanais 

Academic Editor

PLOS ONE